# Combining Explicit and Implicit Regularization for Efficient Learning in Deep Networks

**Dan Zhao**
New York University
`dz1158@nyu.edu`

## Abstract

Works on implicit regularization have studied gradient trajectories during the optimization process to explain why deep networks favor certain kinds of solutions over others. In deep linear networks, it has been shown that gradient descent implicitly regularizes toward low-rank solutions on matrix completion/factorization tasks. Adding depth not only improves performance on these tasks but also acts as an accelerative pre-conditioning that further enhances this bias towards low-rankedness. Inspired by this, we propose an explicit penalty to mirror this implicit bias which only takes effect with certain adaptive gradient optimizers (e.g. Adam). This combination can enable a degenerate single-layer network to achieve low-rank approximations with generalization error comparable to deep linear networks, making depth no longer necessary for learning. The single-layer network also performs competitively or out-performs various approaches for matrix completion over a range of parameter and data regimes despite its simplicity. Together with an optimizer's inductive bias, our findings suggest that explicit regularization can play a role in designing different, desirable forms of regularization and that a more nuanced understanding of this interplay may be necessary.

## 1   Introduction

Much work has poured into understanding why and how highly over-parameterized, deep neural networks with more parameters than training examples generalize so effectively despite long-held notions to the contrary [12, 13]. This generalization puzzle has only deepened as deep learning models often generalize well simply by optimizing its training error on an under-determined problem.

To explain this, previous works have focused on how gradient-based optimization induces an implicit bias on optimization trajectories, particularly in deep (i.e., over-parameterized) settings, tending towards solutions with certain properties [7, 27] like those that generalize well. In contrast, while explicit regularization has seen wide-spread usage in various settings (e.g. weight decay, dropout [60]), its role in explaining generalization has been less certain given its inability to prevent over-fitting on random labels [70] or its absence in deep models that generalize well on their own.

Some works have focused on a simple test-bed to formalize and isolate the mechanisms through which implicit regularization operates—namely, *matrix completion*. Given some observed subset of an unknown, low-rank matrix $W^\star$, the task is to recover the unseen entries. A key observation [27] has been how gradient descent on a shallow linear neural network, with sufficiently small learning rate and near-zero initialization, pushes towards low-rank solutions on its own. This has led to the conjecture [27] that gradient descent induces implicit regularization that minimizes the nuclear norm.

This conjecture has been put into doubt by work [7] showing that gradient descent not only promotes low-rank solutions in the shallow case on matrix completion tasks, but its implicit regularization is further strengthened with increased depth—deep linear neural networks [7]) are able to produce

36th Conference on Neural Information Processing Systems (NeurIPS 2022).

solutions with lower rank and more accurate completion than those from minimizing the nuclear norm. Others [6] have also shown how increased depth or over-parameterization can provide an accelerative pre-conditioning to regression problems for faster convergence in deep linear networks.

**Contributions** Despite these findings, some questions still remain:

- Are there explicit norm-based penalties that can mirror the effect of implicit regularization so as to better harness this phenomenon for improved or more efficient learning? Previous work [7] has conjectured that implicit regularization cannot be characterized by explicit norm-based penalties, but whether these penalties can produce similar effects is unclear.
- Do implicit and explicit forms of regularization interact in any meaningful way? Can we modify the implicit biases of optimizers with explicit regularizers so as to promote better kinds of performance? Some work [11] has begun to draw inspiration from implicit regularization to create explicit regularizers, but their interactions are less clear.
- Previous works [6, 7] have shown that depth can act as a powerful pre-conditioning to accelerate convergence or enhance implicit tendencies towards certain simpler or well-generalizing solutions. Can this effect be produced without depth?

To try and shed more light on these questions, we propose an explicit penalty that takes the ratio between the nuclear norm of a matrix and its Frobenius norm ($\|W\|_\star / \|W\|_F$) and study its effects on the task of matrix completion. This penalty can be interpreted as an adaptation of the Hoyer measure [32] to the spectral domain or as a particular normalization of the nuclear-norm penalty that is commonly used to proxy for rank in convex relaxations of the problem.

Studying implicit regularization can be difficult as it is not always possible to account for all other sources of implicit regularization. For a more precise study of the implicit biases of optimizers and their interactions with our penalty, we use matrix completion and deep linear networks as tractable, yet expressive, and well-understood test-beds that admit a crisp formulation of the mechanism through which implicit regularization operates [7, 28]. In particular, we show the following:

1. A depth 1 linear neural network (i.e., a degenerate network without any depth) trained with this penalty can produce the same rank reduction and deliver comparable, if not better, generalization performance than a deep linear network—all the while converging faster. In short, depth is no longer necessary for learning.

2. The above result only occurs under Adam and, to some extent, its close variants. This suggests that different optimizers, each with their own inductive biases, can interact differently with explicit regularizers to modify dynamics and promote certain solutions over others.

3. With the penalty, we achieve comparable or better generalization and rank-reduction performance against various other techniques (Fig. 4) even in low data regimes (i.e., fewer observed entries during training) where other approaches may have no recovery guarantees.

4. Furthermore, the penalty under Adam enables linear neural networks of all depth levels to produce similar well-generalizing low-rank solutions largely independent of depth, exhibiting a degree of *depth invariance*.

In this specific case, it appears that the learning dynamics which occur through the inductive bias of depth can be compressed or replaced with the right combination of optimization algorithm and explicit penalty. These properties may make deep linear networks more efficient for a variety of related matrix completion, estimation, or factorization tasks, ranging from applications in efficient reinforcement learning [58] to adversarial robustness [68], NLP [1], and others.

Previous conjectures and works [6, 7, 70] have largely dismissed the necessity of explicit regularization in understanding generalization in deep learning, leaving its overall role unclear. Similarly, despite its relative popularity, Adam and other adaptive optimizers have received their share of doubt [61, 62] regarding their effectiveness in producing desirable solutions. Our results suggest a subtle but perhaps important interplay between the choice of the optimizer, its inductive bias, and the explicit penalty in producing different optimization dynamics and, hence, different kinds of solutions. If so, then in these interactions, explicit regularization may yet have a role to play.

**Paper Overview** In Section 2, we review previous work. Section 3 details our experiments and findings. In Section 4, we extend our experiments to a common real-world benchmark and compare our method with other methods. We conclude in Section 5 with a discussion on future work.

## 2 Related Work

Implicit regularization has been studied extensively with recent work focusing on optimization trajectories in deep, over-parameterized settings [6, 7, 10]. One avenue in particular has focused on linear neural networks (LNNs) [26, 29, 37, 57] given their relative tractability and the similarity of their learning dynamics to those of non-linear networks. In settings with multiple optima, [28] has shown how LNNs trained on separable data can converge with an implicit bias to the max margin solution. Others [7] have demonstrated how gradient flow converges towards low-rank solutions where depth acts as an accelerative pre-conditioning [6]. In [3], for natural and vanilla gradient descent (GD), different kinds of pre-conditioning are shown to impact bias-variance and risk trade-offs in over-parameterized linear regression, but this is less clear for adaptive gradient optimizers.

Building upon notions of acceleration and pre-conditioning dating back to Nesterov [50] and Newton, Adam's [39] effectiveness—and its close variants [24, 44, 45]—in optimizing deep networks faster makes clear the importance of adaptive pre-conditioning. Though some [61, 62] have doubted their effectiveness due to potential issues that can harm generalization, others [31, 42, 67, 71] have demonstrated advantages from their adaptive preconditioning; despite speeding up optimization, however, the effect of pre-conditioning on generalization has been less clear as some [2, 23, 65] have argued that the "sharpness" of minima achieved can vary depending on the choice of optimizer. More recent works have characterized pre-conditioning and mechanisms of implicit regularization around "edges of stability" for optimizers where the training regime occurs within a certain sharpness threshold, defined as the maximum eigenvalue of the loss Hessian [8, 21, 22].

Matrix completion and factorization have themselves long been an important focus for areas like signal recovery [17] and recommendation systems [14] from theoretical bounds for recovery and convergence [18, 19, 46, 54] to practical algorithms and implementations [34, 47]. We refer to [20] for a comprehensive survey. These tasks and related ones have also served as test-beds for understanding implicit regularization; [7, 27] use matrix factorization and sensing to study gradient flow's implicit bias towards low-rank solutions, conjecturing that algorithmic regularization may not correspond to minimization of any norm-based penalty. [64] studies the implicit bias of mirror descent on matrix sensing, showing that the solution interpolates between the nuclear and Frobenius norms depending on the underlying matrix. In a related thread, [43] has shown that gradient flow with any commuting parameterization is equivalent to continuous mirror descent with a specific Legendre function, generalizing previous results [9, 27, 63] that have characterized the implicit bias of GD.

Interestingly, [11] illustrates how the discrete steps of GD can regularize optimization trajectories away from large gradients towards flatter minima, developing an explicit regularizer to embody and reinforce this implicit effect directly. To our knowledge, our paper is the first to propose a ratio penalty in matrix completion to study the interplay between explicit and implicit regularization.

## 3 Experiments and Findings

### 3.1 Setup

Formally, we have a ground-truth matrix $W^\star \in \mathbb{R}^{m \times n}$ whose observed entries are indexed by the set $\Omega$. We define the projection $\mathcal{P}_\Omega(W^\star)$ to be a $m \times n$ matrix such that the entries with indices in $\Omega$ remain while the rest are masked with zeros. We are interested in the following optimization:

$$\min_W \mathcal{L}(W) \coloneqq \min_W \|\mathcal{P}_\Omega(W^\star) - \mathcal{P}_\Omega(W)\|_F^2 + \lambda R(W) \quad (1)$$

where $\|\cdot\|_F$ is the Frobenius norm, $R(\cdot)$ is an explicit penalty, and $\lambda \geq 0$ is the tuning parameter. While we consider various penalties, our main focus is demonstrating the effects of our proposed penalty $R(W) = \|W\|_*/\|W\|_F$. Following earlier works [4], we define a *deep linear neural network* (DLNN) through the following over-parameterization, or deep factorization, of $W$:

$$W = W_N W_{N-1} \ldots W_1 \quad (2)$$

under the loss function in (1) where $W_i \in \mathbb{R}^{d_i \times d_{i-1}}, i \in \{1, \ldots, N\}$ denotes the weight matrix corresponding to depth $i$ or the $i$-th layer. Here, $N$ denotes the depth of the network/factorization where $N = 2$ corresponds to *matrix factorization* or a shallow network, $N \geq 3$ corresponds to *deep matrix factorization* or a deep network, and $N = 1$ is the degenerate case (no depth/factorization).

We refer to the matrix $W$, the product of the $N$ weight matrices in Eq. (2), as the *end-product matrix* as per [7]. As such, the end-product matrix $W$ is the solution produced in estimating $W^\star$ or, conveniently, the DLNN itself.

In our analyses, we focus on rank 5 matrices as the ground truth $W^*$ and parameterize our DLNN $W$ with $d_0 = \ldots = d_N = m = n = 100$ (i.e., weight matrices $W_i \in \mathbb{R}^{100 \times 100}$, $\forall i$) for illustrative purposes, but our results extend to other ranks (e.g. see Appendix A.2). We follow previous work [7] and employ the effective rank [56] of a matrix to track and quantify the rank of $W$ in our experiments, defined as: e-rank$(W) = \exp\{H(p_1, \ldots, p_n)\}$ where $H$ is the Shannon entropy, $p_i = \sigma_i / \|\sigma\|_1$, $\{\sigma_i\}$ are the unsigned singular values of $W$, and $\|\cdot\|_1$ is the $\ell_1$ norm. The numerical instability of the numeric rank measure is a known issue [56], resulting in unreliable and unstable rank estimates. We leave a detailed discussion of experiment settings to Appendix A.1.

## 3.2 Depth, without penalty

We first establish a baseline by characterizing the inductive biases of un-regularized gradient descent and un-regularized Adam to better understand their dynamics in the presence of our penalty.

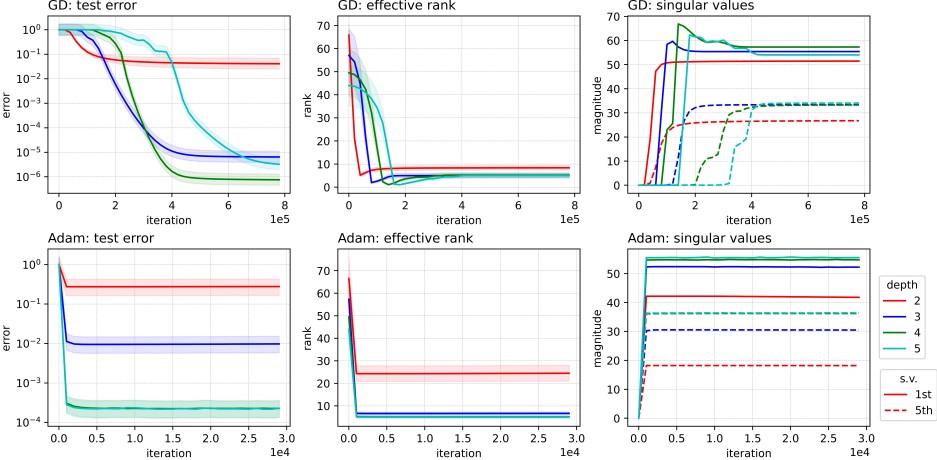

**Figure 1:** Dynamics of *un-regularized gradient descent (GD) and Adam*. Plots show the performance of GD over networks of depths 2/3/4/5 for rank 5 matrices of size $100 \times 100$. Colors correspond to different depth levels and shaded regions correspond to error bands. The left column depicts generalization error as a function of depth and training iterations. The middle column depicts the change in effective rank across depths and over training iterations. The right column shows the 1st and 5th largest singular values for each depth across training iterations. For singular values, a solid line indicates the 1st largest singular value while a dotted line indicates the 5th largest within each depth level (colored lines). We omit the remaining singular values to avoid clutter.

**Gradient Descent** Previous work [7] has shown that depth enhances gradient descent's implicit regularization towards low rank, characterized by the following trajectories on the end-product matrix $W$ and its singular values $\{\sigma_i\}$ (for details, see [6, 7]):

$$\dot{\sigma}_i = -N(\sigma_i(t)^2)^{\frac{N-1}{N}} \cdot \mathbf{u}_i^\top \nabla_W \mathcal{L}(W(t)) \mathbf{v}_i \tag{3}$$

$$\text{vec}(\dot{W}) = -P_W \text{vec}\left(\nabla_W \mathcal{L}(W)\right) \tag{4}$$

where $\dot{\sigma}_i$ is the time derivative of $\sigma_i(t)$, the $i$-th singular value of $W(t)$, $\{\mathbf{u}_i, \mathbf{v}_i\}$ are the left and right singular vectors of $W(t)$ corresponding to $\sigma_i(t)$, $N$ is the network's depth, $\nabla_W \mathcal{L}(W(t))$ is the loss gradient with respect to the end-product matrix $W$ at time $t$, $\text{vec}(\cdot)$ denotes (column-first order) vectorization, $\dot{W} = dW/dt$ is the time evolution of the end-product matrix or (equivalently) the DLNN itself, $P_W = \sum_{j=1}^{N} (W^\top W)^{\frac{N-j}{N}} \otimes (WW^\top)^{\frac{j-1}{N}}$, and $\otimes$ denotes the Kronecker product. We suppress the explicit dependency on $t$ for simplicity and note that full dynamics in Eq. (3) require non-degeneracy (non-trivial depth, $N > 1$); otherwise, they reduce to just $\mathbf{u}_i^\top \nabla_W \mathcal{L}(W(t)) \mathbf{v}_i$.

In Eq. (4), $P_W$ can be seen as a pre-conditioning onto the gradient that, with sufficient depth ($N \geq 2$), accelerates movements already taken in the optimization [6]. As depth/over-parameterization increases, this acceleration intensifies while larger singular values and their movements become more

pronounced than their smaller counterparts, driving singular value separation and a decrease in rank of the recovered matrix (Fig. 1 top row). The singular values evolve at uneven paces depending on the depth; increasing depth increases the gap in the time it takes between the 1st and 5th largest singular values to develop while also taking longer to stabilize. These effects are even more pronounced when comparing the five largest singular values to the remaining ones. Only with sufficient depth ($N > 2$) do solutions produced by un-penalized gradient descent minimize rank so as to recover the rank of the underlying matrix and produce solutions with low test error.

**Adam**  Analyzing Adam can be difficult given its exponentially moving average of gradients; to simplify our analysis, we borrow some assumptions from [5] to approximate Adam's dynamics via gradient flow by assuming that the discounted gradients can be well-approximated by their expectation. (see Appendix A.4 for more details).

**Theorem 1.** *Under the assumptions above and of [7], the trajectory of the singular values $\sigma_i$ of the end-product matrix $W$ can be approximately characterized as:*

$$\dot{\sigma}_i = -\text{vec}(\mathbf{v}_i \mathbf{u}_i^\top)^\top P_{W,G} \text{vec}(\nabla_W \mathcal{L}(W)) \tag{5}$$

*Similarly, the trajectory of the end-product matrix $W$ itself can be approximately characterized as:*

$$\text{vec}(\dot{W}) = -P_{W,G} \text{vec}(\nabla_W \mathcal{L}(W)) \tag{6}$$

*where $P_{W,G} = \sum_{j=1}^{N}((WW^\top)^{\frac{j-1}{N}} \otimes (W^\top W)^{\frac{N-j}{N}})G_j$ is p.s.d. and $G_j$ is a diagonal matrix for layers $j \in \{1, \ldots, N\}$. Specifically, $G_j = \text{diag}(\text{vec}(S_j))$, $[S_j]_{m,n} = [(\nabla_{W_j}\mathcal{L}(W)^2 + s_j^2)^{-1/2}]_{m,n}$, $\nabla_{W_j}\mathcal{L}(W) = \partial\mathcal{L}(W)/\partial W_j$ is layer $j$'s loss gradient, and $s_j^2 = \text{var}(\nabla_{W_j}\mathcal{L}(W))$.*

*Proof.* See Appendix A.4. □

Via this approximation, the pre-conditioning induced by Adam can be characterized as a modification of gradient descent's $P_W$, which now normalizes each layer by the square-root of its squared layer-wise loss gradient $(\partial\mathcal{L}(W)/\partial W_j)^2$ and the gradient variance $s_j^2$, before summing across all depth levels. Unlike before, the variance of the loss gradient comes into play. Whereas before the pre-conditioning served as a purely accelerative effect that intensifies with depth, its normalization by the gradient variance of each layer $W_j$ can now either dampen or further accelerate the trajectory.

Empirically, we see that depth enhances the implicit bias towards low-rank solutions for both Adam and gradient descent albeit differently (Fig. 1, middle column); in deeper regimes ($N > 2$), Adam minimizes rank to exact/near-exact rank recovery more smoothly than gradient descent via faster ($10^4$ vs. $10^5$ iterations) and more uniform convergence (Fig. 1, bottom row). With Adam, singular value dynamics exhibit significantly more uniform evolution regardless of depth in contrast to gradient descent (Fig. 1, right), leading to different types of trajectories and solutions.

### 3.3   Depth, with penalty

**Gradient Descent**  We now characterize the dynamics of gradient flow with our penalty.

**Theorem 2.** *Under the assumptions of [6], the evolution of the singular values of the end-product matrix, under gradient descent with the penalty, can be approximated in the following fashion:*

$$\dot{\sigma}_r = \dot{\sigma}_r^{GF} - \frac{\lambda N}{||W||_F^2}\left(1 - \frac{||W||_*}{||W||_F}\right)\sigma_r^{\frac{3N-2}{2}} \tag{7}$$

*where $\lambda \geq 0$ is the regularization parameter and $\dot{\sigma}_r^{GF}$ denotes the un-regularized singular value trajectory under gradient flow in Eq. (3). Similarly, the evolution of $W$ can be approximated via:*

$$\text{vec}(\dot{W}) = -P_W\left(\text{vec}\left(\nabla_W \mathcal{L}(W)\right) + \lambda\frac{\text{vec}(UV^\top - U\tilde{\Sigma}V^\top)}{||W||_F^2}\right) \tag{8}$$

*where $U, V$ contain the left and right singular vectors of $W$, $P_W$ is the pre-conditioning in Eq. (4), and $\tilde{\Sigma} = \frac{||W||_*}{||W||_F}\Sigma$ where $\Sigma$ contains the singular values of $W$ along its diagonal.*

*Proof.* See Appendix A.4 for details. □

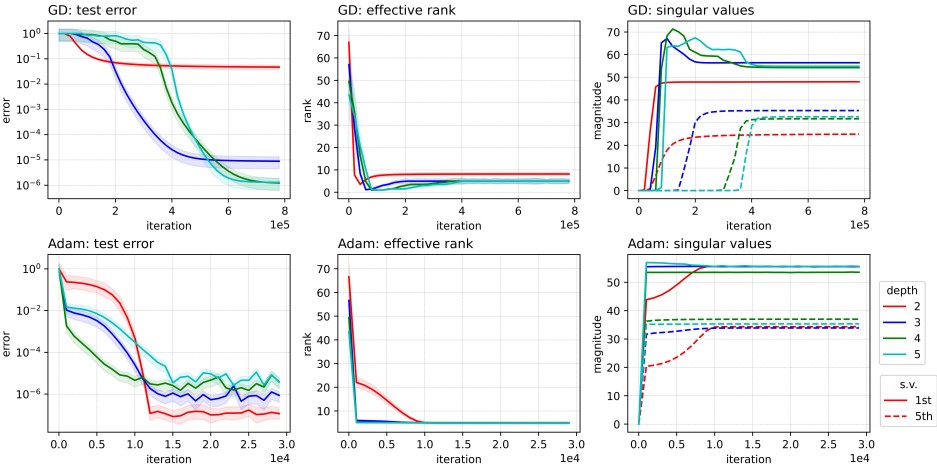

**Figure 2:** Dynamics of *regularized gradient descent (GD) and regularized Adam* with our penalty. Plots above show the performance of GD with our proposed penalty over networks of depths 2/3/4/5 for rank 5 matrix completion. Setup is identical to Fig. 3. Here, $\lambda = 10^{-4}$ but results hold for a range of values $\lambda \in [10^{-4}, 10^{-1}]$.

The penalty can be seen as an additive component to the original trajectory in Eq. (3) that also intensifies with increased depth, making movements in larger singular values more pronounced than those of smaller ones. As a result, it can enable more pronounced singular value separation than before ($\frac{2N-1}{N}$ vs. $\frac{3N-2}{2}$), depending on $\lambda$. Increasing depth continues to push down rank but, unlike before (Eq. (3), Eq. (4)), the penalty now allows singular value trajectories to depend on their own magnitudes even without depth (Eq. (7) with $N = 1$), providing an additional degree of freedom. The penalty also allows each singular value to depend on its relative weight within the distribution of singular values through $(1 - ||W||_* / ||W||_F)$ rather than just its own absolute magnitude.

In Eq. (8), we also see that the depth-dependent accelerative pre-conditioning $P_W$ now acts on a new term: while the first term can be interpreted as the typical influence of reducing the loss via training and gradient optimization on the network's trajectory, the new term can be interpreted as a spectral-based component that can be used by $P_W$ to further enhance the spectral trajectory of $W$ at higher depths, like in Eq. (7). Looking at the diagonals, the new term can be seen as a spectrally re-scaled version of $W$ that influences its trajectory in a way that accounts for each singular value's weight relative to its entire spectrum: $\frac{\mathbf{u}_i^\top \mathbf{v}_i}{||W||_F^2}(1 - \frac{||W||_*}{||W||_F}\sigma_i)$.

Empirically, comparing the un-regularized case (Fig. 1 top row) to the regularized case (Fig. 2 top row), we see that the penalty helps increase the speed at which rank is reduced, inducing faster rates of rank reduction earlier in training. Unlike in un-regularized gradient descent where deeper networks take longer to exhibit rank reduction, the penalty enables near simultaneous reductions in rank across all depths (Fig. 2 top row, middle), making it less dependent on depth.

**Adam** With Adam, the penalty's effect differs considerably in terms of the solutions produced.

**Theorem 3.** *Under the same assumptions, with the proposed penalty, the evolution of the end-product matrix and its singular values can be approximated via the following:*

$$\dot{\sigma} = -\text{vec}(\mathbf{v}_i \mathbf{u}_i^\top)^\top P_{W,G}\left(\text{vec}\left(\nabla_W \mathcal{L}(W)\right) + \lambda \frac{\text{vec}(UV^\top - U\tilde{\Sigma}V^\top)}{||W||_F^2}\right)$$

$$\text{vec}(\dot{W}) = -P_{W,G}\left(\text{vec}\left(\nabla_W \mathcal{L}(W)\right) + \lambda \frac{\text{vec}(UV^\top - U\tilde{\Sigma}V^\top)}{||W||_F^2}\right) \tag{9}$$

*Proof.* Follows from Eq. (5) and Eq. (6) with our penalty. See Appendix A.4 for details. □

Empirically, we note a new development: there is a large degree of *depth invariance* as rank is pushed down and low test error is achieved almost independently of depth (Fig. 2, bottom row), even at depth 2 (i.e., a shallow network). Exact rank recovery of the underlying matrix is now possible at all depths, unlike gradient descent, and the networks converge to solutions faster by an order of magnitude.

From a shallow network ($N = 2$), increasing the depth does not induce any material changes in the solutions produced as the penalized DLNN under Adam produces low-rank solutions that achieve exact rank recovery and low test error faster and better than previous settings.

Moreover, we see that this combination of Adam with the penalty also retains some of the accelerative effect of Adam. Specifically, we see more favorable generalization properties and smoother development of singular values whose convergence speed is at least an order of magnitude faster than under gradient descent ($10^4$ vs. $10^5$ iterations)—whose effects do not vary much with depth. As the singular values evolve, we see that their paths are relatively robust to depth, exhibiting near-identical behavior with significantly less dependence on depth than before (Fig. 2, right most column).

### 3.4 No depth, with penalty

Given the beneficial depth-invariant effects produced by the penalty under Adam in both deep ($N > 2$) and shallow ($N = 2$) networks, we now consider its effects in a limiting case: a degenerate depth 1 network (i.e., no depth; $N = 1$). It is known [70] that gradient descent in this setting converges to the minimum Frobenius ($\ell_2$) norm solution, which does not necessarily induce low-rankedness. As expected, training a depth 1 network with gradient descent fails to produce a well-generalizing or low-rank solution (Fig. 3, top row) as learning is heavily constrained without depth.

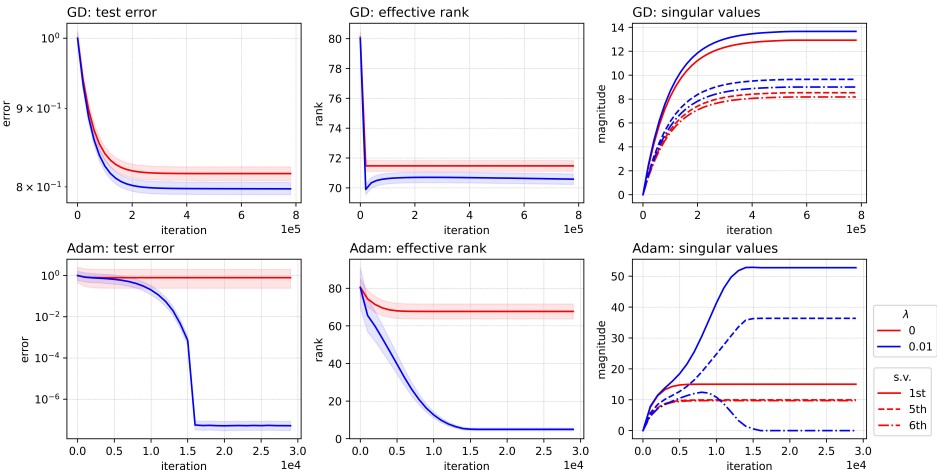

**Figure 3:** Performance comparison between choice of optimization and regularizer in a depth 1 network. Top row corresponds to gradient descent and bottom corresponds to Adam. Note that $\lambda = 0$ (red line) corresponds to the un-regularized setting. Here, $\lambda = 10^{-2}$ but results hold for values $\lambda \in [10^{-6}, 10^{-1}]$.

Yet, despite being ineffective under gradient descent, the penalty is again effective under Adam (Fig. 3, bottom row) even without depth, generalizing as well as if not better than deep networks ($N > 2$). We note that replacing our penalty with other proxies of rank or spectral sparsity, like the nuclear norm, does not work (Table 1). As described earlier, under un-regularized gradient flow with no depth, network dynamics collapse as singular value trajectories reduce to $\mathbf{u}_i^\top \nabla_W \mathcal{L}(W(t)) \mathbf{v}_i$ and the depth dependent accelerative pre-conditioning vanishes ($P_W = I_{mn}$). We see this empirically (e.g. Fig. 3 top row and Table 1) as solutions from un-regularized gradient descent generalize poorly and fail at rank recovery. In contrast, a depth 1 network trained under Adam with the penalty not only achieves low test error (Fig. 3, bottom-left), but also recovers the underlying rank of the ground truth matrix—behaving qualitatively like a deep network. As such, we see that the depth invariance of Adam and the penalty in deeper networks also extends to the case of a depth-less degenerate network.

Without depth, a key component that appears to help the network learn under Adam and the penalty is its variance-weighted gradient term $\nabla_W \mathcal{L}(W) \cdot G$, as defined in Eq. (6), along with the term $P_{W,G}\left(\lambda \frac{\text{vec}(UV^\top - U\tilde{\Sigma}V^\top)}{||W||_F^2}\right)$ which reduces to $G\left(\lambda \frac{\text{vec}(UV^\top - U\tilde{\Sigma}V^\top)}{||W||_F^2}\right)$ without depth. Interestingly, the variance of the loss gradient and the ratio $\eta^2 = \text{var}(\nabla_W \mathcal{L}(W))/\nabla_W \mathcal{L}(W)^2$ formed from $\nabla_W \mathcal{L}(W) \cdot G$ resembles an inverse signal-to-noise ratio that both have come up in other works as

important quantities that are strongly predictive of generalization capabilities [35] or are essential to finding optimal variance adaption factors for loss gradients [5].

## 3.5 Comparison with other penalties and optimizers

**Table 1:** Results for rank 5 matrix completion across various optimizer/penalty/depth combinations in terms of test error (**Err**) and effective rank (**Rk**, rounded to nearest integer) of the estimated matrix. **Ratio** denotes our ratio penalty ($||\cdot||_*/||\cdot||_F$), **Sch p:q** denotes the ratio of two Schatten (quasi)norms ($||\cdot||_{S_p}/||\cdot||_{S_q}$) as penalty, **Nuc** denotes the nuclear norm penalty, **None** is no penalty, and $a \cdot e\, b$ denotes $a \cdot 10^b$. Best results—in terms of both absolute test error (lower the better) and rank reduction (closer to 5 the better) as well as depth invariance in terms of error and rank—are in bold. For more results, see Appendix A.3.

| Optimizer | Depth | Ratio | | Sch $\frac{1}{2}:\frac{2}{3}$ | | Sch $\frac{1}{3}:\frac{2}{3}$ | | Sch $\frac{1}{3}:\frac{1}{2}$ | | Nuc | | None | |
|---|---|---|---|---|---|---|---|---|---|---|---|---|---|
| | | Err | Rk | Err | Rk | Err | Rk | Err | Rk | Err | Rk | Err | Rk |
| Adam [39] | 1 | **4e-7** | **5** | 0.72 | 33 | 0.80 | 45 | 0.81 | 53 | 0.36 | 6 | 1.00 | 79 |
| | 3 | **4e-7** | **5** | 3e-6 | 5 | 1e-5 | 5 | 6e-6 | 5 | 0.30 | 5 | 0.04 | 6 |
| Adagrad [25] | 1 | 0.58 | 31 | 0.81 | 60 | 0.97 | 32 | 0.79 | 60 | 0.12 | 8 | 0.80 | 70 |
| | 3 | 3e-7 | 5 | 9e-7 | 5 | 1e-5 | 5 | 2e-7 | 5 | 0.05 | 6 | 4e-3 | 6 |
| Adamax [39] | 1 | **4e-7** | **5** | 0.76 | 44 | 0.85 | 22 | 0.80 | 58 | 0.05 | 6 | 0.81 | 72 |
| | 3 | **7e-7** | **5** | 3e-6 | 5 | 7e+5 | 1 | 6e-6 | 5 | 0.07 | 7 | 0.01 | 7 |
| RMSProp | 1 | 2e-4 | 6 | 0.08 | 4 | 1.6e+3 | 5 | 1.8e+3 | 8 | 0.05 | 8 | 0.80 | 70 |
| | 3 | 0.03 | 5 | 8e-4 | 5 | 2e-3 | 5 | 1.9 | 5 | 0.05 | 6 | 0.11 | 14 |
| GD | 1 | 0.81 | 67 | 0.81 | 62 | 0.80 | 47 | 0.81 | 60 | 0.82 | 59 | 0.83 | 72 |
| | 3 | 0.51 | 3 | 0.25 | 5 | 0.56 | 3 | 0.39 | 5 | 0.24 | 4 | 1e-5 | 5 |

**Other Combinations** Our results do not preclude the possibility that other optimizers and penalties can produce similar or better effects. For completeness, and inspired by the properties of our penalty (e.g. ratio-like, spectral, non-convex), we experiment with various optimizers and penalties (Table 1) to compare their interactions across shallow ($N = 1$) and deep ($N = 3$) settings. We note that our ratio penalty under Adam and its close variant Adamax, both initially proposed in [39], are the only combinations that largely show depth invariance in test error and rank recovery whereas others require depth to reduce rank or generalize better. Though the nuclear norm exhibits some depth invariance, it is unable to find well-generalizing solutions and fails in rank reduction under gradient descent where depth is still necessary. Other combinations also fail in enabling a depth 1 network to perform as well as deeper ones. Due to space constraints, we leave a fuller treatment and characterization of each optimizer's inductive bias and its interaction with different regularizers for future work.

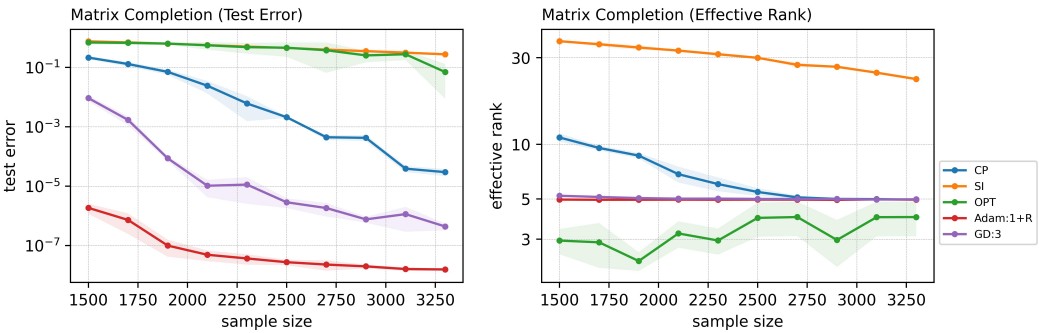

**Figure 4:** Comparative performance in test error and rank minimization for rank 5 matrix completion. $x$-axis stands for the number of observed entries ($\mathbb{R}^{100 \times 100}$, so out of $100 \times 100 = 10^4$ entries) and shaded regions indicate error bands. **Adam:1+R** refers to a depth 1 network trained with Adam and penalty, **CP** is the minimum nuclear norm solution, **GD:3** is a depth 3 network trained with gradient descent, **OPT** is `OptSpace` [38], and **SI** is `SoftImpute` [47]. To reduce clutter, we omit results with similar performance as **GD:3** (e.g. GD:4, GD:5).

**Comparative Performance** We also note our penalty's comparative performance (Fig. 4) against other methodologies for matrix completion across various sample sizes (i.e., the amount of observed entries, uniformly sampled, made available for training). A depth 1 network with Adam and the penalty (Fig. 4, **Adam:1+R**, red line) outperforms all other methods including an un-regularized

DLNN in terms of test error and degree of rank compression/recovery across varying sample sizes. Even at lower sample sizes, the depth 1 network generalizes better than methods such as `SoftImpute` [47], `OptSpace` [38], the minimum nuclear norm solution [19], and DLNNs trained with gradient descent ($N \geq 3$) by at least an order of magnitude. It also outperforms other methods across various data regimes from small sample sizes to large ones, improving as the sample size grows.

# 4   Results on real data

**Table 2:** Performance evaluations on MovieLens100K. Results are categorized by model, whether additional data or features (e.g. external/side information, implicit feedback, graph features, etc.) beyond the explicit ratings in the interaction matrix are used, and test error as measured by root mean squared error (RMSE, lower is better) on different splits in **(a)** and **(b)**. Since various approaches use different train-test proportions, results [16, 53] on two common splits are included. Results from using Adam with the penalty are in bold.

| Model | Uses side info, add. features, or other info, etc? | 90% RMSE |
|---|---|---|
| Depth 1 LNN | No | |
|   w. GD | | 2.814 |
|   w. GD+penalty | | 2.808 |
|   w. Adam | | 1.844 |
|   **w. Adam+penalty** | | **0.915** |
| User-Item Embedding | No | |
|   w. GD | | 2.453 |
|   w. GD+penalty | | 2.535 |
|   w. Adam | | 1.282 |
|   **w. Adam+penalty** | | **0.906** |
| NMF [48] | No | 0.958 |
| PMF [48] | No | 0.952 |
| SVD++ [41] | Yes | 0.913 |
| NFM [30] | No | 0.910 |
| FM [55] | No | 0.909 |
| GraphRec [53] | No | 0.898 |
| AutoSVD++ [59] | Yes | 0.904 |
| GraphRec+sidefeat.[53] | Yes | 0.899 |
| GraphRec+graph/side feat.[53] | Yes | 0.883 |

**(a)** Performance on 90:10 (90%) train-test split

| Model | Uses side info, add. features, or other info, etc? | 80% RMSE |
|---|---|---|
| Depth 1 LNN | No | |
|   w. GD | | 2.797 |
|   w. GD+penalty | | 2.821 |
|   w. Adam | | 1.822 |
|   **w. Adam+penalty** | | **0.921** |
| User-Item Embedding | No | |
|   w. GD | | 2.532 |
|   w. GD+penalty | | 2.519 |
|   w. Adam | | 1.348 |
|   **w. Adam+penalty** | | **0.919** |
| IMC [33, 66] | Yes | 1.653 |
| GMC [36] | Yes | 0.996 |
| MC [18] | Yes | 0.973 |
| GRALS [52] | Yes | 0.945 |
| sRGCNN (sRMGCNN) [49] | Yes | 0.929 |
| GC-MC [16] | Yes | 0.910 |
| GC-MC+side feat. [16] | Yes | 0.905 |

**(b)** Performance on 80:20 (80%) train-test split

Lastly, a natural question might be: how well do our results extend to real-world data? To answer this, we consider MovieLens100K [15]—a common benchmark used to evaluate different approaches for recommendation systems. It consists of ratings from 944 users on 1,683 movies, forming an interaction matrix $M \in \mathbb{R}^{944 \times 1683}$ where the goal is to predict the rest of $M$ after observing a subset.

Unlike our earlier experiments, the values here are discrete in the range $\{1, 2, 3, 4, 5\}$ and $M$ is of high, or near full, rank. Given these differences and more conventional empirical approaches in recommendation systems, we apply our penalty in two ways. The first way is as before: training a depth 1 network with Adam and the penalty (Depth 1 LNN, Table 2). The second way is to impose our penalty on a classic user-item embedding model (User-Item Embedding, Table 2 [40]) that combines user-specific and item-specific biases with a dot product between a latent user and latent item embedding; we apply our penalty separately on either the item or the user embedding layer. Though approaches solely utilizing explicit ratings have fallen out of favor in lieu of ones incorporating additional information and complex designs (e.g. graph-based, implicit feedback, deep non-linear networks), we nonetheless evaluate the effects of our penalty within this simple framework. We compare the results from these two approaches with a variety of approaches that use specialized architectures, deep non-linear networks, additional side information, etc., beyond $M$.

From Table 2, we see that Adam and the penalty (**w. Adam+penalty**) can improve performance over the baseline of gradient descent (GD) or Adam alone. Surprisingly, a depth 1 network with Adam and the penalty can outperform or come close to other more specialized approaches despite its simplicity; however, in contrast to the other methods, it does so without any specialized or additional architectures (e.g. helper models/networks), external information beyond $M$ (e.g. implicit feedback,

side information), construction of new graphs or features, non-linearites, higher-order interactions (e.g. factorization machines), or—for the depth 1 case—even any factorization at all. More precise tuning (e.g. better initialization, learning schedules) or usage of other information/features may yield further improvements on this task and others that involve matrix completion or factorization [1, 58, 68]. We leave these for fuller evaluation and further study in future work.

## 5 Discussion

The dynamics of optimization trajectories—induced together by the choice of optimizer, parameterization, loss function, and architecture—can play an important role in the solutions produced and their ability to generalize. Depth and gradient descent-like algorithms have been key ingredients to the successes of deep learning. On matrix completion/factorization, the proposed penalty helps produce well-generalizing solutions and perfect rank recovery even with a degenerate depth 1, or depth-less, network. Does that mean our penalty, together with Adam's own inductive bias, is producing an effect similar to implicit regularization under gradient descent with depth, but better?

We suspect not. While we concur with the conjecture in [7]—namely, a reductionist view which suggests that implicit regularization can be entirely encapsulated by an explicit norm-based penalty is likely an incorrect over-simplification—we believe that there is merit in studying both implicit and explicit forms of regularization to examine their interplay. Our work suggests that we may be able to partially replicate the successes of *deep* learning by selectively combining optimization methods with explicit penalties via better model design or encoding of inductive biases, but this remains unclear.

Many questions remain open from our limited analyses which, due to space considerations, we leave for future work. For instance, how well do lessons from DLNNs translate to their non-linear counterparts or other tasks (e.g. classification)? How does this relate to learning regimes with larger learning rates or discrete trajectories (i.e., beyond gradient flow)? A more rigorous analysis of the properties (e.g. convergence) of Adam, adaptive gradient methods, and other optimizers in the presence of explicit regularizers may better our understanding. It remains unclear whether implicit regularization is a bug or a feature in deep over-parameterized networks. Nonetheless, our findings suggest the possibility that it can be harnessed and transformed to desirable effect.

## Acknowledgments and Disclosure of Funding

We thank Kellin Pelrine for his feedback and Derek Feng for his help with a previous collaboration on which this work builds. We also thank the anonymous reviewers for their valuable comments and feedback throughout the process.

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
