# OpenReview forum: "Combining Explicit and Implicit Regularization for Efficient Learning in Deep Networks"
_NeurIPS.cc/2022/Conference — NeurIPS 2022 Accept_

### Official Review · Reviewer_C4mW · 2022-06-14

**Rating:** 7
**Confidence:** 3
**Soundness:** 4 excellent
**Presentation:** 4 excellent
**Contribution:** 3 good

**Summary:**

The authors develop a novel regularization method that improves the performance of shallow linear neural networks on tasks that can be represented as low-rank matrix completion/factorization. They offer theoretical results for analyzing such models under vanilla gradient descent vs. ADAM, and show that an adaptive scheme allows shallower networks to converge to better solutions.

**Questions:**

Either make each lemma a theorem / corollary, or explicitly state the overarching theorem.

Line 168-9: It would be helpful to characterize the condition under which normalization will dampen or accelerate the trajectory. Why does it accelerate in the case you specified?

Line 230-1: "This similarly holds for un-regularized Adam" is confusing after reading the previous sentence, and suggests the opposite of what you want to say. Reword / swap the order.

Eqn (6): GD -> GF

I think the appendix should include explicit proofs of Lemmas 2 and 3. Even if they are easy to compute, it will still help some readers follow along more easily.

Add bolding to Table 1 to highlight that your penalty is superior for all the adaptive optimizers.

**Limitations:**

The authors acknowledge that the connection between linear and nonlinear networks in this context is not explored, and that the gradient flow regime may not hold in some practical settings. The authors point to good future directions.

**Strengths And Weaknesses:**

Strengths

The proposed method is simple and effective, supported by thorough theoretical and empirical justification.
The experiments are logical and comprehensive (including appendix), with good interesting results that are easy to interpret.
The text is clear and well-written. The figures are also clear.


Weaknesses

In general, it would be helpful to be even more explicit about the connection between the forms of the various dynamics equations and the empirical results. For example, it would be nice to plot the predicted trajectories of the singular values against the actual trajectories. I still don't have good intuition for the behavior of the dynamics equations, so highlighting different regimes of behaviors would be helpful. For example, when the authors say that a term may accelerate convergence (line 168-9, 202), it would be helpful to describe (or give intuition for) the conditions under which such acceleration occurs, before jumping to the empirical demonstrations that acceleration is indeed observed. How does the convergence rate or error scale with effective rank of the underlying matrix (Appendix experiment does not really answer this)?

As the authors acknowledge in limitations, the scope of their findings is fairly narrow. Even though this is a theory paper, perhaps the motivation could be strengthened by noting that shallow networks are extremely fast and lightweight and thus could find use in applications where these attributes are important.

The title would suggest a very different line of work than what is being examined. Indeed, it is a little misleading in the sense that the paper's main results pertain to depth-1 and depth-2 linear networks, whereas the title writes "deep networks". IMO the title should highlight that the authors develop a novel regularization method that is effective for shallow networks. The title also suggests that the authors design a novel implicit regularization scheme, whereas it seems that implicit regularization refers to an existing property of gradient-based methods (the tendency to favor low-norm solutions). It seems that this implicit regularization is innate to both SGD and Adam, but it would be clearer if the authors instead highlight the property of adaptive optimizers that makes them more effective in the regime studied.

---

> ### Author Response · Authors · 2022-08-02
> **Re: reviewer comments**
>
> Thank you for your comments and feedback, we really appreciate them. Please see our responses below.
> - We completely agree that a more rigorous analysis in understanding different regimes (what conditions accelerate/deccelerate) along the path to convergence, given different parameter settings, is essential.  Unfortunately, we don’t focus on deriving convergence rates explicitly and focus more on characterizing dynamics, which is a limiting factor in the scope of our paper. We have ongoing work due to the space limits in this paper with a slightly different focus on this topic, building off of recent work of [1] but with depth 1 and our characterizations of Adam via gradient flow and at depth 1. This ongoing work will hopefully allow us to better answer this question.
>
> [1] "A convergence analysis of gradient descent for deep linear neural networks" by Arora, Cohen, et al. (2019)
>
> As the authors acknowledge in limitations, the scope of their findings is fairly narrow. Even though this is a theory paper, perhaps the motivation could be strengthened by noting that shallow networks are extremely fast and lightweight and thus could find use in applications where these attributes are important.**
> - Thank you bringing this good point up, we have included a note on this in the introduction.
>
> The title would suggest a very different line of work than what is being examined. Indeed, it is a little misleading in the sense that the paper's main results pertain to depth-1 and depth-2 linear networks, whereas the title writes "deep networks". IMO the title should highlight that the authors develop a novel regularization method that is effective for shallow networks. The title also suggests that the authors design a novel implicit regularization scheme, whereas it seems that implicit regularization refers to an existing property of gradient-based methods (the tendency to favor low-norm solutions). It seems that this implicit regularization is innate to both SGD and Adam, but it would be clearer if the authors instead highlight the property of adaptive optimizers that makes them more effective in the regime studied.
> - Thank you for this point; if submission requirements allow us to change the title, we hope to modify the title to better reflect this point.
>
> **Questions:**
>
> - Either make each lemma a theorem / corollary, or explicitly state the overarching theorem.
>     - Thank you. We have revised some of our mathematical presentation in the main body of the paper and reorganized the singular value and end-product matrix dynamics, placing each pair of each subsection into its own theorem.
>
> - Line 230-1: "This similarly holds for un-regularized Adam" is confusing after reading the previous sentence, and suggests the opposite of what you want to say. Reword / swap the order.
>     - Thank you for this point — we have corrected this so as to not introduce confusion on this front.
>
> - Eqn (6): GD -> GF
>     - Thank you for pointing this out — we use gradient descent (GD) and gradient flow (GF) interchangably but should have been more careful in exercising discretion between when to use, and precisely mean, GD vs. GF. We have made this correction and similar ones throughout the paper.
>
> - I think the appendix should include explicit proofs of Lemmas 2 and 3. Even if they are easy to compute, it will still help some readers follow along more easily.
>     - Thank you for making this point. We have now included material in the appendix (A.5) on deriving these expressions.
>
> - Add bolding to Table 1 to highlight that your penalty is superior for all the adaptive optimizers.
>     - Thank you — we have bolded the relevant parts of Table 1 as suggested.

---

> > ### Comment · Reviewer_C4mW · 2022-08-04
> > **Great work**
> >
> > Thanks authors for your response, I enjoyed reading the paper. This work is very solid and I look forward to reading your future work on the convergence rates.

---

> > > ### Author Response · Authors · 2022-08-08
> > > **Thank you**
> > >
> > > Thank you for your comments, questions, and overall feedback on the paper -- we very much appreciate them.

---

### Official Review · Reviewer_LKjw · 2022-07-11

**Rating:** 8
**Confidence:** 4
**Soundness:** 4 excellent
**Presentation:** 4 excellent
**Contribution:** 4 excellent

**Summary:**

This paper proposes a penalty for gradient descent that consists of the ratio between the nuclear (trace) norm of the weight matrix and its Frobenius norm. The paper shows analytically and empirically, in the task of matrix completion, that linear networks trained with the proposed penalty exhibit a large degree of independence on depth, as long as the models are optimised with Adam (or Adamax), but not with vanilla gradient descent. In particular, the paper presents results that show that in the degenerate case, that is a 1-layer linear network, the model trained with Adam and the proposed penalty achieves generalisation similar to deep networks as well as low rank and spectral sparsity.

**Questions:**

* Question: While the main claim is that the proposed penalty induces depth invariance with Adam, in Figure 2 we see some differences in the test error dynamics (bottom left) across different depths. In particular, with 2 layers, the dynamics are qualitatively very different and the model achieves lower error than with more layers. What do the authors think this is due to?
* Suggestion: include repetitions in the experimental setup in order to obtain confidence intervals for the experimental analysis.
* Suggestion: the actual proposed penalty is only expressed in the middle of a paragraph in Section 1. I think it would be clearer to include the explicit equation of the optimisation (Eq. 1) with the penalty. This would make it easier for future readers.
* Suggestion: the title of Section 3.5, "Comparative performance" seems quite general. I would consider making it more specific, such as "Comparison with other penalties and optimisers".
* Suggestion: same goes for the title of Section 4. How about "Results on real-world data"?

**Limitations:**

As mention above, the authors do briefly but explicitly discuss the limitations of their paper.

**Strengths And Weaknesses:**

This is, to the best of my judgement, an overall very good paper. First of all, the paper makes a very concrete contribution: the proposal of a penalty for gradient descent. The authors present a set of experiments that analyse 1) baseline conditions without the penalty, 2) the impact of the penalty on both vanilla gradient descent and models optimised with Adam, with particular attention to 3) the degenerate case (depth 1). Furthermore, the paper extends the empirical analysis by studying other optimisers and penalties, as well as some experiments on a real world data set (movies recommendations). With this experimental setup, the authors find an interesting result, namely that the proposed penalty induces certain degree of depth independence when the models are optimised with Adam but not with natural gradient descent. The paper provides a theoretical analysis of the dynamics under certain common assumptions and shows empirical results that analyse the generalisation performance, the rank of the weight matrix and the dynamics of the 1st and 5th singular values.

In terms of significance, I believe these are interesting results that can encourage further research into understanding the optimisation dynamics of neural networks and the interplay between implicit and explicit regularisation, which are topics that have attracted significant interest in the last decade. In terms of originality, the authors claim that the proposed penalty is novel and I am myself not aware of previous work proposing it, though I am not sufficiently up to date with the most recent literature on this specific domain.

The paper is very well written, much better than the average submission I have reviewed at machine learning conferences. The introduction is very clear and it smoothly motivates the problem by summarising the relevant related works and gaps in the literature. Section 2, on related work is easy to follow and concise. Then, the core section of the paper, Section 3, is very reasonably structured, following a logical order (setup, baseline models without penalty, models with the penalty, the degenerate case and other optimisers and penalties. The paper ends with a brief section where the role of the penalty is analysed on a real world data set and a discussion. I would also like to note that I have not identified any typo, which is extremely rare in the papers I have reviewed.

Regarding quality, I think the paper is technically sound, the analyses seem correct to my judgement (though I have not thoroughly reviewed all the supplementary material) and the results support the claims. Furthermore, I would like to highlight that the authors humbly discuss the significance of their paper, in light of what the paper does show and does not show, as well as the limitations and aspects that are left for future work. This is, as well, and unfortunately, rare in machine learning submissions, but I highly appreciate it.

I can identify some weaknesses, but these are minor and most are actually mentioned by the authors in Section 5. These mostly have to do with the limited scope of the experiments and analyses, which leave some open questions, such as: the role of the penalty on non-linear networks, on other tasks and data sets, whether other penalties can achieve the same depth-independence, a more thorough analysis of why the penalty only works with Adam and whether the same dynamics could be induced with a modification of the penalty on vanilla gradient descent. Since it is impossible to address all these questions properly in a single paper and the authors actually mention them, I consider these weaknesses minor, or not at all.

Finally, one aspect that could be improved in the paper is the fact that the results (plots) from the empirical analyses seem to reflect single experiments. I think the conclusions and insights would be stronger if repetitions were carried out (with various initialisation seeds, for instance, and, ideally, choices of $\Omega$) in order to obtain a measure of the confidence intervals for the dynamics.

---

> ### Author Response · Authors · 2022-08-02
> **re: reviewer feedback**
>
> Thank you for your evaluation and we are glad you enjoyed our paper, we really appreciate your comments and feedback. We’ve made attempts to address them in the time given and detail our responses to each of your points below.
>
>
> **Questions:**
>
> - While the main claim is that the proposed penalty induces depth invariance with Adam, in Figure 2 we see some differences in the test error dynamics (bottom left) across different depths. In particular, with 2 layers, the dynamics are qualitatively very different and the model achieves lower error than with more layers. What do the authors think this is due to?
>     - This is a very good question. One hypothesis has been the strength of regularization -- although a large range of regularization parameter values work in achieving the end result of lower error and rank recovery, it is unclear if deeper networks require differing regularization strengths of our penalty than shallower ones -- deeper networks may be more susceptible to higher variance in the propagation of gradients/values so that at depth 2, the regularization strength used for these experiments may have been well-suited by higher/lower values may be better suited for shallower/deeper networks/layers. This question may receive better insight had our paper also focused more on deriving convergence rates of deep linear networks (as a function of depth, rank of underlying matrix, observation size, and regularization strength + our penalty), which unfortunately is a limiting factor in the scope of this paper but is a focus of a paper in progress. Another hypothesis is that of a potential regime shift; going from a shallow model (depths 1, 2 both show similar loss curve trajectories compared to those of depth 3+) to a deeper model (depth > 2) may unlock additional degrees of freedom in terms of model expressiveness. However, this is unclear to us based upon the experiments and work in this paper.
>
> - Suggestion: include repetitions in the experimental setup in order to obtain confidence intervals for the experimental analysis.
>     - Thank you for bringing up this point — we have included error bands for our experimental analyses in our plots/diagrams in our experimental section. We’ve also made efforts to improve the aesthetics of our plots/diagrams in order to improve readability and visibility. While we included these for the error curves and rank curves, we left them out for singular values due to the fact that their magnitudes fluctuate greatly depending on initialization (as what ultimately matters is the relative distribution of their magnitudes rather than their absolute magnitudes), potentially adding more clutter/confusion than insight into the plots with the large error bars.
>
> - Suggestion: the actual proposed penalty is only expressed in the middle of a paragraph in Section 1. I think it would be clearer to include the explicit equation of the optimisation (Eq. 1) with the penalty. This would make it easier for future readers.
>     - Thank you for this point — looking back, the explicit form of the proposed penalty only appeared once in the main body of the paper. We have now also mentioned the penalty explicitly in section 3.1, experimental setup, and also in the appendix when referring to it.
>
> - Suggestion: the title of Section 3.5, "Comparative performance" seems quite general. I would consider making it more specific, such as "Comparison with other penalties and optimisers".
>     - Thank you for the suggestion — indeed, the title of section 3.5 was a bit generic and we have changed the title of Section 3.5 to better reflect the contents and purpose of this section.
> - Suggestion: same goes for the title of Section 4. How about "Results on real-world data"?
>     - Thank you  — we have changed the title of section 4 as well to better reflect its contents and convey them better to the reader.

---

> > ### Author Response · Authors · 2022-08-09
> > **Re: depth 1/2 dynamics vs. depth 3+ dynamics**
> >
> > We have also finished conducting some very preliminary experiments re: the first question and report some of the initial findings here in case it helps shed more light on the question. As mentioned, the plots in our paper all showed results where $\lambda$ was set to the same value (0.01) for proper comparisons across optimizers/depths; as discussed, we see that depth 1 and depth 2 networks, with the penalty, tend to have a slightly different error curve trajectory relative to depth 3+ networks with penalty. However, as we decrease $\lambda$ for depths 1 and 2, we see that this causes them to behave more like their deeper counterparts, increasing the average number of iterations that they need to arrive at test error levels of around $10^{-6}$, closer to the average number of iterations that their deeper counterparts take to get to around the same level. While this lends some credence to our hypothesis described in our earlier comment above, a more rigorous analysis is clearly still needed to verify/understand this "trade-off" and the reasons behind it.

---

> > > ### Comment · Reviewer_LKjw · 2022-08-09
> > > **Response to authors 2**
> > >
> > > Thank you for your explanations about this observation and for looking deeper into the possible reasons. I think the two hypothesis are reasonable and I agree that this would need a more in-depth analysis in future work.

---

> > ### Comment · Reviewer_LKjw · 2022-08-09
> > **Response to authors 1**
> >
> > Thank you for answering my questions and considering the suggestions. In particular, I very positively value the inclusion of multiple runs per experiment and the update of the figures with error bands. I believe this should make a more convincing and solid paper.

---

### Official Review · Reviewer_ngkq · 2022-07-11

**Rating:** 6
**Confidence:** 3
**Soundness:** 3 good
**Presentation:** 3 good
**Contribution:** 3 good

**Summary:**

Deep Linear Neural Networks gradient descent implicitly regularizes towards low-rank solutions in the matrix completion/factorization tasks. This work proposes an explicit penalty term that mimics the rank minimization behavior of deep linear networks. The proposed method is invariant to depth. A depth 1 linear neural network equipped with the proposed penalty produces the same rank reduction with comparable performance as deep matrix factorization. The authors note that their proposed penalty works really well with Adam optimizer but not with Gradient Descent. They speculate that the observed behavior could be due to the interaction of inductive biases inherent to the optimizers with explicit regularizers. They conduct a series of experiments to demonstrate this difference between Adam and Gradient Descent. Finally, they run experiments on real world data - MovieLens100k and show that they perform competitively against the existing methods which often use specialized architecture, parametrization, additional side information, etc.

**Questions:**

If the authors could address the following weaknesses, I'd be more than happy to increase the scores -
* The proposed method is depth invariant. If the authors could list potential areas where this could be useful, it would immensely help the paper.
* It would be great if the authors could throw some theoretical insight into the reason behind performance discrepancy between various optimizers.

**Ethics Review Area:**

["I don’t know"]

**Limitations:**

The authors have adequately addressed the limitations and potential negative societal impact of their work.

**Strengths And Weaknesses:**

Strengths:
* This work takes a unique approach towards rank minimization. They propose an explicit penalty that mirrors the implicit rank minimization of deep matrix factorization.
* The proposed method is depth invariant which is an interesting property.
* The proposed method achieves better generalization and rank-reduction performance beyond many techniques even in low data regimes.
*  The authors note that their proposed penalty works really well with Adam optimizer but not with other optimizers such as Gradient Descent. The authors have done sufficient empirical evaluations to demonstrate the differences between the behavior of the penalty term under different optimizers.

Weaknesses:
* The proposed method is depth invariant. If the authors could list potential areas where this could be useful, it would immensely help the paper.
* It would be great if the authors could throw some theoretical insight into the reason behind performance discrepancy between various optimizers.

---

> ### Author Response · Authors · 2022-08-02
> **Re: reviewer comments**
>
> Thank you very much for your time and feedback, we really appreciate it.
>
> 1. The proposed method is depth invariant. If the authors could list potential areas where this could be useful, it would immensely help the paper.
>
>     * The most immediate benefit is likely to be efficiency, especially in terms of training/inference. In resource or compute constrained settings such as on embedded devices etc, this is an important point since this potentially allows for the training of a much more simpler network than a heavily over-parameterized one while overall preserving performance; similarly, smaller networks also mean smaller memory requirements for storing network weights/architecture, etc which, together with other methodologies, can also help lead to improvements in compressing or distilling deep networks (for instance, works like [1] rely upon the task we study in our paper, i.e. low rank matrix factorization, for online embedding compression for NLP tasks). Many other areas also rely on some form of low-rank matrix estimation/completion/factorization, from reinforcement learning [2] to adversarial robustness [3] just to name a few, but some cases requires performance on par with a deep network/factorization; with this depth invariance/penalty, we can potentilaly simplify this greatly to a 1 layer for improved efficiency without significnatly sacrificing performance. We hope that future work, including a paper that we are currently working on, will also focus on deep nonlinear networks in order to expand the scope of applicability--we have seen some success thus far with ReLU non-linearities applied to our networks but more work is required to expand the scope in more generality.
>
>     [1] "Online Embedding Compression for Text Classification using Low Rank Matrix Factorization" (2019) Acharya, Goel, et al. AAAI 2019
>
>     [2] "ME-Net: Towards Effective Adversarial Robustness with Matrix Estimation" (2019) Yang, Zhang, et al. ICML 2019
>
>     [3] "Sample Efficient Reinforcement Learning via Low-Rank Matrix Estimation" (2020) Shah, Song et al. NeurIPS 2020
>
>    *  Another related benefit potentially concerns hyper-parameter optimization; in deep networks, whenever different techniques are used (e.g. layer norm, stochastic neuron/layer dropout, etc.), there's a need to perform some extent of hyperparameter sweep over different configurations for these techniques, which can grow with the number of layers (i.e. depth) of a network. Our approach may also be able to help save computational effort on this end as well if it allows a one layer network to perform as effectively as a deeper one.
>
>    * A second area which would require more thought and investigation would be model interpretability and explainability; instead of a deep network with each layer capturing/processing/learning something different, an equivalent one layer linear model can help make dissecting and interpreting latent factors/features of a task more easy and efficient.  While linear networks and variants are still in use for things like recommendation systems, matrix sensing applications, etc., an important extension would be to non-linear settings -- if this could be possible, then depth invariance can take on more usefulness.
>
> 2. It would be great if the authors could throw some theoretical insight into the reason behind performance discrepancy between various optimizers.
>    * This is a very good question which we unfortunately do not have a definitive answer to. While we have some preliminary hypotheses, they have not been verified and due to the space considerations of this paper we felt it best not to include any preliminary work given the paper's main focus/content. Part of the motivation for our extensive experimentation with different penalty/optimizer combinations is exactly this, seeing if empirical simulations can help tease out noticeable patterns or trends. In this paper, what we have observed is that adaptive gradient methods like Adam, with the penalty, are the only ones producing these beneficial effects.
>    * Looking closely, Adam and Adamax achieve this effect the best (compared to RAdam, NAdam, and other less related optimizers) likely due to the fact that, in our approximate characterizations, they produce a noise-to-signal term (a function of the gradient variance over squared gradient magnitude) that the loss gradient is inversely weighted by, which may help gradient updates in more informative directions. For other optimizers, through the same approximation we make via gradient flow, the end dynamics no longer includes this term by itself and is impacted by additional components (from the differences in optimizers), which can be seen as biasing the gradient variance and/or noise-to-signal term. In some optimizers, the update rule do not provide unbiased estimates of the first/second moments of the gradient.  We agree that a more focused treatment across other optimizers is necessary to better understand this discrepancy.

---

### Official Review · Reviewer_kNgp · 2022-07-12

**Rating:** 5
**Confidence:** 3
**Soundness:** 3 good
**Presentation:** 1 poor
**Contribution:** 2 fair

**Summary:**

This paper proposes a new regularization for deep linear networks. The analyses sections showed theoretical analysis empirical results of this regularizer in the context of 1-layer/deep networks and SGD/Adam optimizer in matrix completion tasks where the ground truth matrix is low-rank. The main experiment was conducted on a recommender system task, showing the proposed regularizer on very simple models could perform complicated methods.


**Questions:**

1. Line 45: what does "acceleration" mean? Please specify or give examples here. Also, I don't think this paper later even partially answers this question, except by deferring it to some sort of preconditioning.
1. In the main experiments, are the features produced by linear mapping or nonlinear networks? If it's nonlinear networks, how is the regularization applied?
1. Line 291: what's meant by "without any factorizations"? Isn't the model itself a factorized representation of the data matrix?
1. I see that most derivations aim to express dynamics as preconditioned gradient descent, but could the authors justify why this particular preconditioning is good? Why not any preconditioning?
2. Buttom line of page 19 (For some reason, the line numbers disappear), I cannot think of why the last equality holds. What's the "factorization of Kronecker product"? I did a simple search but could not find any properties related to this line.

**Limitations:**

The discussion mentions some limitations and future work, but should also discuss limitations of the experiment design, including full-batch training, a single matrix factorization task and a single dataset.

**Strengths And Weaknesses:**

## Strengths:
1. I quite like the style of studying simple tasks and the way that the authors interleave illustrative examples with theoretical analyses.
2. The synthetic example is used to compare a wide range of alternative methods/regularizations, clearly showing the advantage of the proposal.
2. The authors performed extensive parameter sweeps as described in the appendix.
3. The authors are clearly technical in their analyses.

## Weaknesses:
1. Writing is unclear to the extent that I cannot clearly see if the authors delivered the promise made in the introduction/contribution paragraph. I'm also not an expert in this field so could just be a background mismatch, but I find that the main message is overcomplicated and partially obscured by very scattered writing.
2. This method does not seem to be directly applicable to nonlinear networks, but the main experiment might have demonstrated this but the description was not clear.
3. The main experiment was only conducted in a single setting and dataset, which is somewhat lacking.
3. The main theoretical advantage seems to be some form of preconditioning, but the authors did not go deep enough to suggest why such preconditioning may be beneficial, apart from being pos-semidef.
3. Mathematical presentation needs to be thoroughly tidied up.
4. The synthetic experiments are trained under full-batch, which does not really produce stochastic gradients.

## Details:
1. Line 54: depth is not necessary compared to deep linear networks, not any networks. I also find this statement and many other statements very strong, and then realize that the authors probably mean in the context of deep linear networks.
2. Lines 30-31 say the task is to "recover the unseen entries", then in 58-62 the author describes that their regularizaton has an effect on the "rank and generalization", and the figure clearly does not give errors on how well the model recovers unseen entries. What exactly is the goal of the task? Is it to recover the unseen entries? To get the correct rank? Or test prediction which I assume is generalization? Please be clear here and everywhere.
3. Lines 120-121: This part describes how the student network weights are set up. The first time I read it, I thought the authors meant that they focused on rank-5 matrices in each layer, but then understood that the rank-5 applies to the teacher matrix $W^*$.
4. 135: what is singular value separation? This is undefined in the paper but gets mentioned a lot later. I'm a non-expert but I think doesn't take an expert to understand, it must be some trends shown in the figures.
5. Line 158: "the trajectory of ... its singular values $\sigma_i$" does not appear in the Lemma. Is this in equation (14) of the Appendix?
6.  I don't think the rates in equations (3) and (6) are ever proved in the appendix.
7. Line 190, comparing figures 1 & 2, top rows, I don't think this statement is supported by the figures. The curves stabilize at mostly some iteration numbers. Further, the green curve in Test Error actually converges slower.
8. Section 4: I have no idea what the authors have done for the recommender system. Please use equations or give a reference.
9. Line 519: the authors should define the rank in the main text explicitly rather than giving a reference in the appendix. If an approximate rank is used, perhaps the authors should also try other possible rank definitions.
10. Line 211: "evolves" -> "evolve"

---

> ### Author Response · Authors · 2022-08-02
> **Re: questions/comments**
>
> Thank you for your thoughtful comments and for pointing out potential areas of improvement. We’ve made attempts to address your points in series below.
> *  We apologize for any confusion caused by our writing or presentation. We have made some adjustments in some of the writing and mathematical presentation to try and ensure the key points come through. In this paper, all references to deep networks are made in relation to deep linear networks unless otherwise stated. We have made some changes throughout to be clearer about the context in which we are making said statements.
>
> *  In terms of the standard setup of matrix completion, the goal is to, in general terms, recover some underlying matrix by observing some portion of it and trying to recover the rest (effectively masking part of the matrix during training and testing the model on the parts it hasn’t seen for training). Typically, an assumption is made (or it is known) that the underlying matrix is low rank—in which case, the ideal goal is to recover the unseen entries as closely as possible (i.e. reduce test error or the error on these test entries based on what the model has learned on training entries) and also produce a solution matrix whose overall rank is low or is close to the rank of the underlying truth matrix.
>
> * As such, the plots in the paper show the test error on these unseen entries and the resultant rank of the recovered matrix. Due to space considerations/limits in the main body of the paper, we have included more explicit clarification on this part of the experimental setup in Appendix A1.
>
> * Re: singular value separation, we apologize for the confusion — we meant, in a less rigorous sense, that singular values become more separated from one another (i.e. larger ones become larger and vice versa due to the compounding as a result of the trajectory). For instance, in a rank 5 matrix, the separation would be the growing distance/difference in magnitude between the top 5 singular values and the remaining values (as seen in the plots of paper).
>
> * As part of tidying up the mathematical presentation of some of the analytical equations, we have now included proofs/references to all equations from the main paper in the appendix. Equation (3) (unregularized gradient flow dynamics) is from previous work which we include a reference to but, due to space considerations, we did not include its proof here.
>
> * Re: the use of rank, we’ve now included a brief discussion in the experiment setup section of the main text. There are several problems with numerical rank, and its variants, which is the reason why previous works have also used the effective rank measure; the numerical rank measure is extremely volatile, providing highly unstable and high-variance estimates of a matrix’s rank.
>
> **Questions:**
> 1. Line 45: what does "acceleration" mean?
>  * By acceleration, we mean both in the sense of faster convergence (i.e. efficient convergence, in terms of rank and error but also to better solutions with closer rank and error to ground truth) as primarily seen in the plots.
>
> 2. In the main experiments, are the features produced by linear mapping or nonlinear networks? If it's nonlinear networks, how is the regularization applied?
> * Our focus in this paper is on deep/shallow linear networks only. We have in limited experiments tried non-linearities, namely ReLU, inserted in-between weight layers and found no qualitative differences. However, other non-linearities may show other substantial differences.
>
> 3. Line 291: what's meant by "without any factorizations"? Isn't the model itself a factorized representation of the data matrix?
> * Thank you for bringing this up and apologies for any confusion here. By this, we mean a depth 1 model -- in section 4, we apply our penalty in one of two ways: to a depth one model (i.e. therefore, without any factorization) and to a more conventional model (akin to a depth 2 "shallow" factorization).
>
> 4. I see that most derivations aim to express dynamics as preconditioned gradient descent, but could the authors justify why this particular preconditioning is good? Why not any preconditioning?
> * To clarify, this pre-conditioning on its own may not be good, but it's the "corrective" or modified effect the penalty has on it that produces a good effect. It is an important open question that we are exploring in more detail in another paper due to space considerations here.
>
> 5. Bottom line of page 19 (For some reason, the line numbers disappear), I cannot think of why the last equality holds. What's the "factorization of Kronecker product"? I did a simple search but could not find any properties related to this line.
> * By factorization of the Kronecker product, we mean the mixed-product property, i.e., given matrices $A_1, A_2, B_1, B_2$ , then  $(A_1 \otimes A_2)(B_1 \otimes B_2) = (A_1B_1) \otimes (A_2 B_2)$ where $\otimes$ denotes the Kronecker product. We’ve corrected and clarified our wording on this  in the appendix.

---

> > ### Author Response · Authors · 2022-08-07
> > **Re: additional comments (due to cut-off from character limit)**
> >
> > We apologize as we've realized some parts of our full response were cut off and left out (likely due to the character limit or something else). After bringing this up with the ACs, we have included the left out parts, and their contexts, below.
> >
> > **Questions**
> >
> > 4.  I see that most derivations aim to express dynamics as preconditioned gradient descent, but could the authors justify why this particular preconditioning is good? Why not any preconditioning?
> >
> > * To clarify, this pre-conditioning on its own may not be good, but it's the "corrective" or modified effect the penalty has on it that produces a good effect. It is an important open question that we are exploring in more detail in another paper due to space considerations here. The key points we hoped to make (both in this paper and with our penalty) is that: (i) although different optimizers (e.g. GD vs Adam) produce similar kinds of implicitly regularizing effects overall with increased depth (i.e. lowering rank and test error) in the setting we consider, they do so in slightly different ways (e.g. differences in the trajectory of error curves, rank curves, and singular value paths), (ii) the use of the "right" kind of explicit regularizers can help or improve the good aspects that modifies the pre-conditoning (e.g. in this case, making it so that these effects are no longer as depth dependent or producing better generalizing solutions/faster rank recovery is why this penalty-modified Adam pre-conditoning is good), and (iii) different explicit regularizers can have different interactions with different optimizers to produce non-trivial dynamics/results.
> >
> > 7. Line 190, comparing figures 1 & 2, top rows, I don't think this statement is supported by the figures. The curves stabilize at mostly some iteration numbers. Further, the green curve in Test Error actually converges slower.
> >
> > * Thank you for bringing this point up. We agree -- what we meant to say was more that there is less variance across depth levels in terms of the development/trajectory of the rank and singular value curves; whereas in fig 1 (top row, GD w/o penalty) in the original submission, we see considerable variance across depth levels in terms of how long (or how many iterations) it takes for the rank to start decreasing or for the top 1/5 singular values to arise/develop depending on the depth of the network, in fig 2 (top row GD w/ penalty), we see that this variance is diminished by quite a bit as the curves across depth levels are more "bunched together" and show less sensitivity to depth level in both how the rank curves decrease across depths but also in the development of 1st/5th largest singular values.
> >
> > 8. Section 4: I have no idea what the authors have done for the recommender system. Please use equations or give a reference.
> >
> > * Thank you for pointing this out; we have included a reference in the real world data experiment section. In line with our response to a related point above, we test our penalty in two ways on the popular movielens100k benchmark: with a depth 1 network (i.e. no matrix factorization) as in our synthetic experiments and with a shallow depth 2 network (i.e. akin to a "typical" matrix factorization with some differences such as the inclusion of user and item biases, embedding layers, etc.), the latter of which is a more typical but dated approach in recommendation systems.
> >
> > 9. Line 519: the authors should define the rank in the main text explicitly rather than giving a reference in the appendix. If an approximate rank is used, perhaps the authors should also try other possible rank definitions.
> >
> > * Thank you for this point -- we have addressed this in one of our bullet-point comments before the Questions section of our response above (i.e. we've included a portion in the beginning of the experimental setup section of the main text clarifying the rank measure we use).
> >
> > 10. Line 211: "evolves" -> "evolve"
> >
> > * Thank you for pointing this out. We have corrected this typo and double-checked the paper for others.

---

> > > ### Comment · Reviewer_kNgp · 2022-08-08
> > > **Thanks for the response**
> > >
> > > Maybe I'm very slow, but could the authors explain this question: Bottom line of page 19 (For some reason, the line numbers disappear), I cannot think of why the last equality holds.

---

> > > > ### Author Response · Authors · 2022-08-09
> > > > **Re: math clarifiation**
> > > >
> > > > No, not at all -- this portion is not very straightforward and is quite involved. We presume you are referring to the last equality right before line 674 near the top of page 21 in the newer rebuttal revision version of the paper (i.e. right after using the mixed-product property of the Kronecker product for matrices $A$ and $B$, and substituting the terms for $A$ and $B$ back in to produce the expression in terms of $W$). Please correct us if not.
> > > >
> > > > Since the derivation is quite involved and due to space/character limits here, we defer to the relevant paper from which we cite the derivation and borrow the results, which can be found in line 678 of the rebuttal revision version of the paper or equivalently [1] (see below) -- specifically Appendix A, pages 11-12 (in the paper version linked below), equations (16) thru to (24) with equations (23) and (24) being the end result we desire (and equations (16) thru (22) describing the derivation). Like we mention throughout our appendix, due to the length of certain proofs we borrow from other works and for the sake of focus and clarity in our paper, we decided to omit reproducing these whole proofs, such as those from [1], and instead make mentions/cite the relevant works to serve as foundations from which we derive our expressions. We hope this helps.
> > > >
> > > > [1] "On the Optimization of Deep Networks: Implicit Acceleration by Overparameterization" Arora, Cohen, and Hazan, ICML 2018 (https://arxiv.org/pdf/1802.06509.pdf)

---

### Meta-Review · Area_Chair_d5x6 · 2022-08-24

**Recommendation:** Accept
**Confidence:** Certain

**Metareview:**

All reviewers recommended that the paper be accepted, and I accordingly recommend the same. I encourage the authors to take into account suggestions made by reviewers so as to further improve the text towards the camera-ready version.

**Award:**

No

---

### Decision · Program_Chairs · 2022-09-14

Accept